# First Order Motion Model for Image Animation

**Aliaksandr Siarohin**
DISI, University of Trento
aliaksandr.siarohin@unitn.it

**Stéphane Lathuilière**
DISI, University of Trento
LTCI, Télécom Paris, Institut polytechnique de Paris
stephane.lathuilire@telecom-paris.fr

**Sergey Tulyakov**
Snap Inc.
stulyakov@snap.com

**Elisa Ricci**
DISI, University of Trento
Fondazione Bruno Kessler
e.ricci@unitn.it

**Nicu Sebe**
DISI, University of Trento
Huawei Technologies Ireland
niculae.sebe@unitn.it

## Abstract

Image animation consists of generating a video sequence so that an object in a source image is animated according to the motion of a driving video. Our framework addresses this problem without using any annotation or prior information about the specific object to animate. Once trained on a set of videos depicting objects of the same category (*e.g.* faces, human bodies), our method can be applied to any object of this class. To achieve this, we decouple appearance and motion information using a self-supervised formulation. To support complex motions, we use a representation consisting of a set of learned keypoints along with their local affine transformations. A generator network models occlusions arising during target motions and combines the appearance extracted from the source image and the motion derived from the driving video. Our framework scores best on diverse benchmarks and on a variety of object categories. Our source code is publicly available[1].

## 1 Introduction

Generating videos by animating objects in still images has countless applications across areas of interest including movie production, photography, and e-commerce. More precisely, image animation refers to the task of automatically synthesizing videos by combining the appearance extracted from a *source image* with motion patterns derived from a *driving video*. For instance, a face image of a certain person can be animated following the facial expressions of another individual (see Fig. 1). In the literature, most methods tackle this problem by assuming strong priors on the object representation (*e.g.* 3D model) [4] and resorting to computer graphics techniques [6, 33]. These approaches can be referred to as *object-specific* methods, as they assume knowledge about the model of the specific object to animate.

Recently, deep generative models have emerged as effective techniques for image animation and video retargeting [2, 41, 3, 42, 27, 28, 37, 40, 31, 21]. In particular, Generative Adversarial Networks (GANs) [14] and Variational Auto-Encoders (VAEs) [20] have been used to transfer facial expressions [37] or motion patterns [3] between human subjects in videos. Nevertheless, these approaches usually rely on pre-trained models in order to extract object-specific representations such as keypoint locations. Unfortunately, these pre-trained models are built using costly ground-truth data annotations [2, 27, 31] and are not available in general for an arbitrary object category. To address this issues, recently Siarohin *et al.* [28] introduced Monkey-Net, the first object-agnostic deep model for image

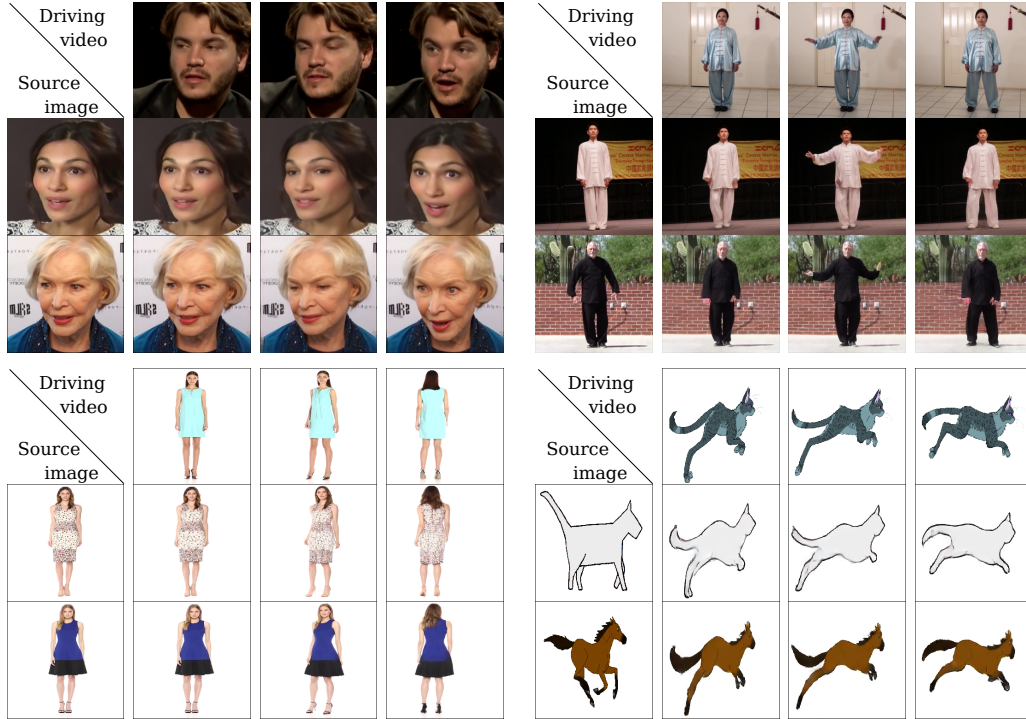

Figure 1: Example animations produced by our method trained on different datasets: *VoxCeleb* [22] (top left), *Tai-Chi-HD* (top right), *Fashion-Videos* [41] (bottom left) and *MGif* [28] (bottom right). We use relative motion transfer for *VoxCeleb* and *Fashion-Videos* and absolute transfer for *MGif* and *Tai-Chi-HD* see Sec. 3.4. Check our project page for more qualitative results[2].

animation. Monkey-Net encodes motion information via keypoints learned in a self-supervised fashion. At test time, the source image is animated according to the corresponding keypoint trajectories estimated in the driving video. The major weakness of Monkey-Net is that it poorly models object appearance transformations in the keypoint neighborhoods assuming a zeroth order model (as we show in Sec. 3.1). This leads to poor generation quality in the case of large object pose changes (see Fig. 4). To tackle this issue, we propose to use a set of self-learned keypoints together with local affine transformations to model complex motions. We therefore call our method a first-order motion model. Second, we introduce an occlusion-aware generator, which adopts an occlusion mask automatically estimated to indicate object parts that are not visible in the source image and that should be inferred from the context. This is especially needed when the driving video contains large motion patterns and occlusions are typical. Third, we extend the equivariance loss commonly used for keypoints detector training [18, 44], to improve the estimation of local affine transformations. Fourth, we experimentally show that our method significantly outperforms state-of-the-art image animation methods and can handle high-resolution datasets where other approaches generally fail. Finally, we release a new high resolution dataset, *Thai-Chi-HD*, which we believe could become a reference benchmark for evaluating frameworks for image animation and video generation.

## 2 Related work

**Video Generation.** Earlier works on deep video generation discussed how spatio-temporal neural networks could render video frames from noise vectors [36, 26]. More recently, several approaches tackled the problem of conditional video generation. For instance, Wang *et al.* [38] combine a recurrent neural network with a VAE in order to generate face videos. Considering a wider range of applications, Tulyakov *et al.* [34] introduced MoCoGAN, a recurrent architecture adversarially trained in order to synthesize videos from noise, categorical labels or static images. Another typical case of conditional generation is the problem of future frame prediction, in which the generated video is conditioned on the initial frame [12, 23, 30, 35, 44]. Note that in this task, realistic predictions can be obtained by simply warping the initial video frame [1, 12, 35]. Our approach is closely related

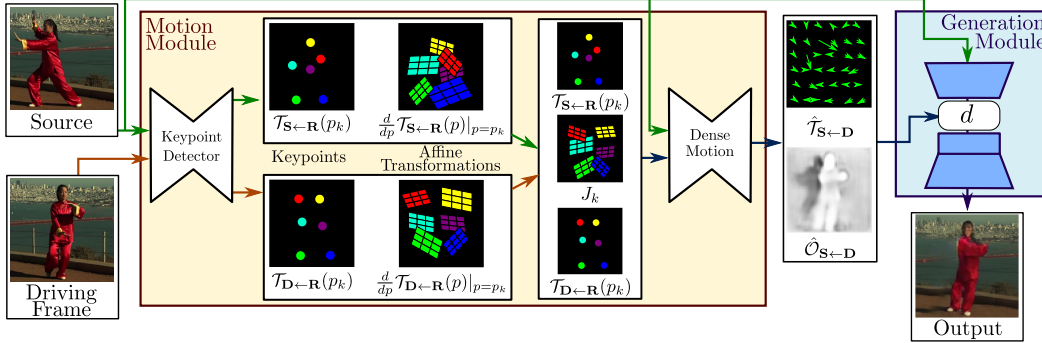

Figure 2: Overview of our approach. Our method assumes a source image $\mathbf{S}$ and a frame of a driving video frame $\mathbf{D}$ as inputs. The unsupervised keypoint detector extracts first order motion representation consisting of sparse keypoints and local affine transformations with respect to the reference frame $\mathbf{R}$. The dense motion network uses the motion representation to generate dense optical flow $\hat{\mathcal{T}}_{\mathbf{S}\leftarrow\mathbf{D}}$ from $\mathbf{D}$ to $\mathbf{S}$ and occlusion map $\hat{\mathcal{O}}_{\mathbf{S}\leftarrow\mathbf{D}}$. The source image and the outputs of the dense motion network are used by the generator to render the target image.

to these previous works since we use a warping formulation to generate video sequences. However, in the case of image animation, the applied spatial deformations are not predicted but given by the driving video.

**Image Animation.** Traditional approaches for image animation and video re-targeting [6, 33, 13] were designed for specific domains such as faces [45, 42], human silhouettes [8, 37, 27] or gestures [31] and required a strong prior of the animated object. For example, in face animation, method of Zollhofer *et al.* [45] produced realistic results at expense of relying on a 3D morphable model of the face. In many applications, however, such models are not available. Image animation can also be treated as a translation problem from one visual domain to another. For instance, Wang *et al.* [37] transferred human motion using the image-to-image translation framework of Isola *et al.* [16]. Similarly, Bansal *et al.* [3] extended conditional GANs by incorporating spatio-temporal cues in order to improve video translation between two given domains. Such approaches in order to animate a single person require hours of videos of that person labelled with semantic information, and therefore have to be retrained for each individual. In contrast to these works, we neither rely on labels, prior information about the animated objects, nor on specific training procedures for each object instance. Furthermore, our approach can be applied to any object within the same category (*e.g.*, faces, human bodies, robot arms etc).

Several approaches were proposed that do not require priors about the object. X2Face [40] uses a dense motion field in order to generate the output video via image warping. Similarly to us they employ a reference pose that is used to obtain a canonical representation of the object. In our formulation, we do not require an explicit reference pose, leading to significantly simpler optimization and improved image quality. Siarohin *et al.* [28] introduced Monkey-Net, a self-supervised framework for animating arbitrary objects by using sparse keypoint trajectories. In this work, we also employ sparse trajectories induced by self-supervised keypoints. However, we model object motion in the neighbourhood of each predicted keypoint by a local affine transformation. Additionally, we explicitly model occlusions in order to indicate to the generator network the image regions that can be generated by warping the source image and the occluded areas that need to be inpainted.

## 3 Method

We are interested in animating an object depicted in a source image $\mathbf{S}$ based on the motion of a similar object in a driving video $\mathcal{D}$. Since direct supervision is not available (pairs of videos in which objects move similarly), we follow a self-supervised strategy inspired from Monkey-Net [28]. For training, we employ a large collection of video sequences containing objects of the same object category. Our model is trained to reconstruct the training videos by combining a single frame and a learned latent representation of the motion in the video. Observing frame pairs, each extracted from the same video, it learns to encode motion as a combination of motion-specific keypoint displacements and local affine transformations. At test time we apply our model to pairs composed of the source image and of each frame of the driving video and perform image animation of the source object.

An overview of our approach is presented in Fig. 2. Our framework is composed of two main modules: the motion estimation module and the image generation module. The purpose of the motion estimation module is to predict a dense motion field from a frame $\mathbf{D} \in \mathbb{R}^{3 \times H \times W}$ of dimension $H \times W$ of the driving video $\mathcal{D}$ to the source frame $\mathbf{S} \in \mathbb{R}^{3 \times H \times W}$. The dense motion field is later used to align the feature maps computed from $\mathbf{S}$ with the object pose in $\mathbf{D}$. The motion field is modeled by a function $\mathcal{T}_{\mathbf{S} \leftarrow \mathbf{D}} : \mathbb{R}^2 \rightarrow \mathbb{R}^2$ that maps each pixel location in $\mathbf{D}$ with its corresponding location in $\mathbf{S}$. $\mathcal{T}_{\mathbf{S} \leftarrow \mathbf{D}}$ is often referred to as backward optical flow. We employ backward optical flow, rather than forward optical flow, since back-warping can be implemented efficiently in a differentiable manner using bilinear sampling [17]. We assume there exists an abstract reference frame $\mathbf{R}$. We independently estimate two transformations: from $\mathbf{R}$ to $\mathbf{S}$ ($\mathcal{T}_{\mathbf{S} \leftarrow \mathbf{R}}$) and from $\mathbf{R}$ to $\mathbf{D}$ ($\mathcal{T}_{\mathbf{D} \leftarrow \mathbf{R}}$). Note that unlike X2Face [40] the reference frame is an abstract concept that cancels out in our derivations later. Therefore it is never explicitly computed and cannot be visualized. This choice allows us to independently process $\mathbf{D}$ and $\mathbf{S}$. This is desired since, at test time the model receives pairs of the source image and driving frames sampled from a different video, which can be very different visually. Instead of directly predicting $\mathcal{T}_{\mathbf{D} \leftarrow \mathbf{R}}$ and $\mathcal{T}_{\mathbf{S} \leftarrow \mathbf{R}}$, the motion estimator module proceeds in two steps.

In the first step, we approximate both transformations from sets of sparse trajectories, obtained by using keypoints learned in a self-supervised way. The locations of the keypoints in $\mathbf{D}$ and $\mathbf{S}$ are separately predicted by an encoder-decoder network. The keypoint representation acts as a bottleneck resulting in a compact motion representation. As shown by Siarohin *et al.* [28], such sparse motion representation is well-suited for animation as at test time, the keypoints of the source image can be moved using the keypoints trajectories in the driving video. We model motion in the neighbourhood of each keypoint using local affine transformations. Compared to using keypoint displacements only, the local affine transformations allow us to model a larger family of transformations. We use Taylor expansion to represent $\mathcal{T}_{\mathbf{D} \leftarrow \mathbf{R}}$ by a set of keypoint locations and affine transformations. To this end, the keypoint detector network outputs keypoint locations as well as the parameters of each affine transformation.

During the second step, a dense motion network combines the local approximations to obtain the resulting dense motion field $\hat{\mathcal{T}}_{\mathbf{S} \leftarrow \mathbf{D}}$. Furthermore, in addition to the dense motion field, this network outputs an occlusion mask $\hat{\mathcal{O}}_{\mathbf{S} \leftarrow \mathbf{D}}$ that indicates which image parts of $\mathbf{D}$ can be reconstructed by warping of the source image and which parts should be inpainted, *i.e.* inferred from the context.

Finally, the generation module renders an image of the source object moving as provided in the driving video. Here, we use a generator network $G$ that warps the source image according to $\hat{\mathcal{T}}_{\mathbf{S} \leftarrow \mathbf{D}}$ and inpaints the image parts that are occluded in the source image. In the following sections we detail each of these step and the training procedure.

### 3.1 Local Affine Transformations for Approximate Motion Description

The motion estimation module estimates the backward optical flow $\mathcal{T}_{\mathbf{S} \leftarrow \mathbf{D}}$ from a driving frame $\mathbf{D}$ to the source frame $\mathbf{S}$. As discussed above, we propose to approximate $\mathcal{T}_{\mathbf{S} \leftarrow \mathbf{D}}$ by its first order Taylor expansion in a neighborhood of the keypoint locations. In the rest of this section, we describe the motivation behind this choice, and detail the proposed approximation of $\mathcal{T}_{\mathbf{S} \leftarrow \mathbf{D}}$.

We assume there exist an abstract reference frame $\mathbf{R}$. Therefore, estimating $\mathcal{T}_{\mathbf{S} \leftarrow \mathbf{D}}$ consists in estimating $\mathcal{T}_{\mathbf{S} \leftarrow \mathbf{R}}$ and $\mathcal{T}_{\mathbf{R} \leftarrow \mathbf{D}}$. Furthermore, given a frame $\mathbf{X}$, we estimate each transformation $\mathcal{T}_{\mathbf{X} \leftarrow \mathbf{R}}$ in the neighbourhood of the learned keypoints. Formally, given a transformation $\mathcal{T}_{\mathbf{X} \leftarrow \mathbf{R}}$, we consider its first order Taylor expansions in $K$ keypoints $p_1, \ldots p_K$. Here, $p_1, \ldots p_K$ denote the coordinates of the keypoints in the reference frame $\mathbf{R}$. Note that for the sake of simplicity in the following the point locations in the reference pose space are all denoted by $p$ while the point locations in the $\mathbf{X}$, $\mathbf{S}$ or $\mathbf{D}$ pose spaces are denoted by $z$. We obtain:

$$\mathcal{T}_{\mathbf{X} \leftarrow \mathbf{R}}(p) = \mathcal{T}_{\mathbf{X} \leftarrow \mathbf{R}}(p_k) + \left( \frac{d}{dp} \mathcal{T}_{\mathbf{X} \leftarrow \mathbf{R}}(p) \Big|_{p=p_k} \right) (p - p_k) + o(\|p - p_k\|), \qquad (1)$$

In this formulation, the motion function $\mathcal{T}_{\mathbf{X} \leftarrow \mathbf{R}}$ is represented by its values in each keypoint $p_k$ and its Jacobians computed in each $p_k$ location:

$$\mathcal{T}_{\mathbf{X} \leftarrow \mathbf{R}}(p) \simeq \left\{ \left\{ \mathcal{T}_{\mathbf{X} \leftarrow \mathbf{R}}(p_1), \frac{d}{dp} \mathcal{T}_{\mathbf{X} \leftarrow \mathbf{R}}(p) \Big|_{p=p_1} \right\}, \ldots \left\{ \mathcal{T}_{\mathbf{X} \leftarrow \mathbf{R}}(p_k), \frac{d}{dp} \mathcal{T}_{\mathbf{X} \leftarrow \mathbf{R}}(p) \Big|_{p=p_K} \right\} \right\}. \quad (2)$$

Furthermore, in order to estimate $\mathcal{T}_{\mathbf{R}\leftarrow\mathbf{X}} = \mathcal{T}_{\mathbf{X}\leftarrow\mathbf{R}}^{-1}$, we assume that $\mathcal{T}_{\mathbf{X}\leftarrow\mathbf{R}}$ is locally bijective in the neighbourhood of each keypoint. We need to estimate $\mathcal{T}_{\mathbf{S}\leftarrow\mathbf{D}}$ near the keypoint $z_k$ in $\mathbf{D}$, given that $z_k$ is the pixel location corresponding to the keypoint location $p_k$ in $\mathbf{R}$. To do so, we first estimate the transformation $\mathcal{T}_{\mathbf{R}\leftarrow\mathbf{D}}$ near the point $z_k$ in the driving frame $\mathbf{D}$, $e.g.$ $p_k = \mathcal{T}_{\mathbf{R}\leftarrow\mathbf{D}}(z_k)$. Then we estimate the transformation $\mathcal{T}_{\mathbf{S}\leftarrow\mathbf{R}}$ near $p_k$ in the reference $\mathbf{R}$. Finally $\mathcal{T}_{\mathbf{S}\leftarrow\mathbf{D}}$ is obtained as follows:

$$\mathcal{T}_{\mathbf{S}\leftarrow\mathbf{D}} = \mathcal{T}_{\mathbf{S}\leftarrow\mathbf{R}} \circ \mathcal{T}_{\mathbf{R}\leftarrow\mathbf{D}} = \mathcal{T}_{\mathbf{S}\leftarrow\mathbf{R}} \circ \mathcal{T}_{\mathbf{D}\leftarrow\mathbf{R}}^{-1}, \tag{3}$$

After computing again the first order Taylor expansion of Eq. (3) (see *Sup. Mat.*), we obtain:

$$\mathcal{T}_{\mathbf{S}\leftarrow\mathbf{D}}(z) \approx \mathcal{T}_{\mathbf{S}\leftarrow\mathbf{R}}(p_k) + J_k(z - \mathcal{T}_{\mathbf{D}\leftarrow\mathbf{R}}(p_k)) \tag{4}$$

with:

$$J_k = \left( \frac{d}{dp} \mathcal{T}_{\mathbf{S}\leftarrow\mathbf{R}}(p) \Big|_{p=p_k} \right) \left( \frac{d}{dp} \mathcal{T}_{\mathbf{D}\leftarrow\mathbf{R}}(p) \Big|_{p=p_k} \right)^{-1} \tag{5}$$

In practice, $\mathcal{T}_{\mathbf{S}\leftarrow\mathbf{R}}(p_k)$ and $\mathcal{T}_{\mathbf{D}\leftarrow\mathbf{R}}(p_k)$ in Eq. (4) are predicted by the keypoint predictor. More precisely, we employ the standard U-Net architecture that estimates $K$ heatmaps, one for each keypoint. The last layer of the decoder uses softmax activations in order to predict heatmaps that can be interpreted as keypoint detection confidence map. Each expected keypoint location is estimated using the average operation as in [28, 24]. Note if we set $J_k = \mathbb{1}$ ($\mathbb{1}$ is $2 \times 2$ identity matrix), we get the motion model of Monkey-Net. Therefore Monkey-Net uses a zeroth-order approximation of $\mathcal{T}_{\mathbf{S}\leftarrow\mathbf{D}}(z) - z$.

For both frames $\mathbf{S}$ and $\mathbf{D}$, the keypoint predictor network also outputs four additional channels for each keypoint. From these channels, we obtain the coefficients of the matrices $\frac{d}{dp} \mathcal{T}_{\mathbf{S}\leftarrow\mathbf{R}}(p)|_{p=p_k}$ and $\frac{d}{dp} \mathcal{T}_{\mathbf{S}\leftarrow\mathbf{R}}(p)|_{p=p_k}$ in Eq. (5) by computing spatial weighted average using as weights the corresponding keypoint confidence map.

**Combining Local Motions.** We employ a convolutional network $P$ to estimate $\hat{\mathcal{T}}_{\mathbf{S}\leftarrow\mathbf{D}}$ from the set of Taylor approximations of $\mathcal{T}_{\mathbf{S}\leftarrow\mathbf{D}}(z)$ in the keypoints and the original source frame $\mathbf{S}$. Importantly, since $\hat{\mathcal{T}}_{\mathbf{S}\leftarrow\mathbf{D}}$ maps each pixel location in $\mathbf{D}$ with its corresponding location in $\mathbf{S}$, the local patterns in $\hat{\mathcal{T}}_{\mathbf{S}\leftarrow\mathbf{D}}$, such as edges or texture, are pixel-to-pixel aligned with $\mathbf{D}$ but not with $\mathbf{S}$. This misalignment issue makes the task harder for the network to predict $\hat{\mathcal{T}}_{\mathbf{S}\leftarrow\mathbf{D}}$ from $\mathbf{S}$. In order to provide inputs already roughly aligned with $\hat{\mathcal{T}}_{\mathbf{S}\leftarrow\mathbf{D}}$, we warp the source frame $\mathbf{S}$ according to local transformations estimated in Eq. (4). Thus, we obtain $K$ transformed images $\mathbf{S}^1, \ldots \mathbf{S}^K$ that are each aligned with $\hat{\mathcal{T}}_{\mathbf{S}\leftarrow\mathbf{D}}$ in the neighbourhood of a keypoint. Importantly, we also consider an additional image $\mathbf{S}^0 = \mathbf{S}$ for the background.

For each keypoint $p_k$ we additionally compute heatmaps $\mathbf{H}_k$ indicating to the dense motion network where each transformation happens. Each $\mathbf{H}_k(z)$ is implemented as the difference of two heatmaps centered in $\mathcal{T}_{\mathbf{D}\leftarrow\mathbf{R}}(p_k)$ and $\mathcal{T}_{\mathbf{S}\leftarrow\mathbf{R}}(p_k)$:

$$\mathbf{H}_k(z) = exp\left( \frac{(\mathcal{T}_{\mathbf{D}\leftarrow\mathbf{R}}(p_k) - z)^2}{\sigma} \right) - exp\left( \frac{(\mathcal{T}_{\mathbf{S}\leftarrow\mathbf{R}}(p_k) - z)^2}{\sigma} \right). \tag{6}$$

In all our experiments, we employ $\sigma = 0.01$ following Jakab *et al.* [18].

The heatmaps $\mathbf{H}_k$ and the transformed images $\mathbf{S}^0, \ldots \mathbf{S}^K$ are concatenated and processed by a U-Net [25]. $\hat{\mathcal{T}}_{\mathbf{S}\leftarrow\mathbf{D}}$ is estimated using a part-based model inspired by Monkey-Net [28]. We assume that an object is composed of $K$ rigid parts and that each part is moved according to Eq. (4). Therefore we estimate $K+1$ masks $\mathbf{M}_k, k = 0, \ldots K$ that indicate where each local transformation holds. The final dense motion prediction $\hat{\mathcal{T}}_{\mathbf{S}\leftarrow\mathbf{D}}(z)$ is given by:

$$\hat{\mathcal{T}}_{\mathbf{S}\leftarrow\mathbf{D}}(z) = \mathbf{M}_0 z + \sum_{k=1}^{K} \mathbf{M}_k \left( \mathcal{T}_{\mathbf{S}\leftarrow\mathbf{R}}(p_k) + J_k(z - \mathcal{T}_{\mathbf{D}\leftarrow\mathbf{R}}(p_k)) \right) \tag{7}$$

Note that, the term $\mathbf{M}_0 z$ is considered in order to model non-moving parts such as background.

## 3.2 Occlusion-aware Image Generation

As mentioned in Sec.3, the source image $\mathbf{S}$ is not pixel-to-pixel aligned with the image to be generated $\hat{\mathbf{D}}$. In order to handle this misalignment, we use a feature warping strategy similar to [29, 28, 15]. More precisely, after two down-sampling convolutional blocks, we obtain a feature map $\boldsymbol{\xi} \in \mathbb{R}^{H' \times W'}$ of dimension $H' \times W'$. We then warp $\boldsymbol{\xi}$ according to $\hat{\mathcal{T}}_{\mathbf{S}\leftarrow\mathbf{D}}$. In the presence of occlusions in $\mathbf{S}$, optical flow may not be sufficient to generate $\hat{\mathbf{D}}$. Indeed, the occluded parts in $\mathbf{S}$ cannot be recovered by image-warping and thus should be inpainted. Consequently, we introduce an occlusion map $\hat{\mathcal{O}}_{\mathbf{S}\leftarrow\mathbf{D}} \in [0, 1]^{H' \times W'}$ to mask out the feature map regions that should be inpainted. Thus, the occlusion mask diminishes the impact of the features corresponding to the occluded parts. The transformed feature map is written as:

$$\boldsymbol{\xi}' = \hat{\mathcal{O}}_{\mathbf{S}\leftarrow\mathbf{D}} \odot f_w(\boldsymbol{\xi}, \hat{\mathcal{T}}_{\mathbf{S}\leftarrow\mathbf{D}}) \tag{8}$$

where $f_w(\cdot, \cdot)$ denotes the back-warping operation and $\odot$ denotes the Hadamard product. We estimate the occlusion mask from our sparse keypoint representation, by adding a channel to the final layer of the dense motion network. Finally, the transformed feature map $\boldsymbol{\xi}'$ is fed to subsequent network layers of the generation module (see *Sup. Mat.*) to render the sought image.

## 3.3 Training Losses

We train our system in an end-to-end fashion combining several losses. First, we use the reconstruction loss based on the perceptual loss of Johnson *et al.* [19] using the pre-trained VGG-19 network as our main driving loss. The loss is based on implementation of Wang *et al.* [37]. With the input driving frame $\mathbf{D}$ and the corresponding reconstructed frame $\hat{\mathbf{D}}$, the reconstruction loss is written as:

$$L_{rec}(\hat{\mathbf{D}}, \mathbf{D}) = \sum_{i=1}^{I} \left| N_i(\hat{\mathbf{D}}) - N_i(\mathbf{D}) \right|, \tag{9}$$

where $N_i(\cdot)$ is the $i^{th}$ channel feature extracted from a specific VGG-19 layer and $I$ is the number of feature channels in this layer. Additionally we propose to use this loss on a number of resolutions, forming a pyramid obtained by down-sampling $\hat{\mathbf{D}}$ and $\mathbf{D}$, similarly to MS-SSIM [39, 32]. The resolutions are $256 \times 256$, $128 \times 128$, $64 \times 64$ and $32 \times 32$. There are 20 loss terms in total.

**Imposing Equivariance Constraint.** Our keypoint predictor does not require any keypoint annotations during training. This may lead to unstable performance. Equivariance constraint is one of the most important factors driving the discovery of unsupervised keypoints [18, 43]. It forces the model to predict consistent keypoints with respect to known geometric transformations. We use thin plate splines deformations as they were previously used in unsupervised keypoint detection [18, 43] and are similar to natural image deformations. Since our motion estimator does not only predict the keypoints, but also the Jacobians, we extend the well-known equivariance loss to additionally include constraints on the Jacobians.

We assume that an image $\mathbf{X}$ undergoes a known spatial deformation $\mathcal{T}_{\mathbf{X}\leftarrow\mathbf{Y}}$. In this case $\mathcal{T}_{\mathbf{X}\leftarrow\mathbf{Y}}$ can be an affine transformation or a thin plane spline deformation. After this deformation we obtain a new image $\mathbf{Y}$. Now by applying our extended motion estimator to both images, we obtain a set of local approximations for $\mathcal{T}_{\mathbf{X}\leftarrow\mathbf{R}}$ and $\mathcal{T}_{\mathbf{Y}\leftarrow\mathbf{R}}$. The standard equivariance constraint writes as:

$$\mathcal{T}_{\mathbf{X}\leftarrow\mathbf{R}} \equiv \mathcal{T}_{\mathbf{X}\leftarrow\mathbf{Y}} \circ \mathcal{T}_{\mathbf{Y}\leftarrow\mathbf{R}} \tag{10}$$

After computing the first order Taylor expansions of both sides, we obtain the following constraints (see derivation details in *Sup. Mat.*):

$$\mathcal{T}_{\mathbf{X}\leftarrow\mathbf{R}}(p_k) \equiv \mathcal{T}_{\mathbf{X}\leftarrow\mathbf{Y}} \circ \mathcal{T}_{\mathbf{Y}\leftarrow\mathbf{R}}(p_k), \tag{11}$$

$$\left( \frac{d}{dp}\mathcal{T}_{\mathbf{X}\leftarrow\mathbf{R}}(p) \bigg|_{p=p_k} \right) \equiv \left( \frac{d}{dp}\mathcal{T}_{\mathbf{X}\leftarrow\mathbf{Y}}(p) \bigg|_{p=\mathcal{T}_{\mathbf{Y}\leftarrow\mathbf{R}}(p_k)} \right) \left( \frac{d}{dp}\mathcal{T}_{\mathbf{Y}\leftarrow\mathbf{R}}(p) \bigg|_{p=p_k} \right), \tag{12}$$

Note that the constraint Eq. (11) is strictly the same as the standard equivariance constraint for the keypoints [18, 43]. During training, we constrain every keypoint location using a simple $L_1$ loss between the two sides of Eq. (11). However, implementing the second constraint from Eq. (12) with

$L_1$ would force the magnitude of the Jacobians to zero and would lead to numerical problems. To this end, we reformulate this constraint in the following way:

$$\mathbb{1} \equiv \left( \frac{d}{dp} \mathcal{T}_{\mathbf{X} \leftarrow \mathbf{R}}(p) \Big|_{p=p_k} \right)^{-1} \left( \frac{d}{dp} \mathcal{T}_{\mathbf{X} \leftarrow \mathbf{Y}}(p) \Big|_{p=\mathcal{T}_{\mathbf{Y} \leftarrow \mathbf{R}}(p_k)} \right) \left( \frac{d}{dp} \mathcal{T}_{\mathbf{Y} \leftarrow \mathbf{R}}(p) \Big|_{p=p_k} \right), \qquad (13)$$

where $\mathbb{1}$ is $2 \times 2$ identity matrix. Then, $L_1$ loss is employed similarly to the keypoint location constraint. Finally, in our preliminary experiments, we observed that our model shows low sensitivity to the relative weights of the reconstruction and the two equivariance losses. Therefore, we use equal loss weights in all our experiments.

### 3.4 Testing Stage: Relative Motion Transfer

At this stage our goal is to animate an object in a source frame $\mathbf{S}_1$ using the driving video $\mathbf{D}_1, \ldots \mathbf{D}_T$. Each frame $\mathbf{D}_t$ is independently processed to obtain $\mathbf{S}_t$. Rather than transferring the motion encoded in $\mathcal{T}_{\mathbf{S}_1 \leftarrow \mathbf{D}_t}(p_k)$ to $\mathbf{S}$, we transfer the relative motion between $\mathbf{D}_1$ and $\mathbf{D}_t$ to $\mathbf{S}_1$. In other words, we apply a transformation $\mathcal{T}_{\mathbf{D}_t \leftarrow \mathbf{D}_1}(p)$ to the neighbourhood of each keypoint $p_k$:

$$\mathcal{T}_{\mathbf{S}_1 \leftarrow \mathbf{S}_t}(z) \approx \mathcal{T}_{\mathbf{S}_1 \leftarrow \mathbf{R}}(p_k) + J_k(z - \mathcal{T}_{\mathbf{S} \leftarrow \mathbf{R}}(p_k) + \mathcal{T}_{\mathbf{D}_1 \leftarrow \mathbf{R}}(p_k) - \mathcal{T}_{\mathbf{D}_t \leftarrow \mathbf{R}}(p_k)) \qquad (14)$$

with

$$J_k = \left( \frac{d}{dp} \mathcal{T}_{\mathbf{D}_1 \leftarrow \mathbf{R}}(p) \Big|_{p=p_k} \right) \left( \frac{d}{dp} \mathcal{T}_{\mathbf{D}_t \leftarrow \mathbf{R}}(p) \Big|_{p=p_k} \right)^{-1} \qquad (15)$$

Detailed mathematical derivations are provided in *Sup. Mat.*. Intuitively, we transform the neighbourhood of each keypoint $p_k$ in $\mathbf{S}_1$ according to its local deformation in the driving video. Indeed, transferring relative motion over absolute coordinates allows to transfer only relevant motion patterns, while preserving global object geometry. Conversely, when transferring absolute coordinates, as in X2Face [40], the generated frame inherits the object proportions of the driving video. It's important to note that one limitation of transferring relative motion is that we need to assume that the objects in $\mathbf{S}_1$ and $\mathbf{D}_1$ have similar poses (see [28]). Without initial rough alignment, Eq. (14) may lead to absolute keypoint locations physically impossible for the object of interest.

## 4 Experiments

**Datasets.** We train and test our method on four different datasets containing various objects. Our model is capable of rendering videos of much higher resolution compared to [28] in all our experiments.

• The *VoxCeleb* dataset [22] is a face dataset of 22496 videos, extracted from YouTube videos. For pre-processing, we extract an initial bounding box in the first video frame. We track this face until it is too far away from the initial position. Then, we crop the video frames using the smallest crop containing all the bounding boxes. The process is repeated until the end of the sequence. We filter out sequences that have resolution lower than $256 \times 256$ and the remaining videos are resized to $256 \times 256$ preserving the aspect ratio. It's important to note that compared to X2Face [40], we obtain more natural videos where faces move freely within the bounding box. Overall, we obtain 12331 training videos and 444 test videos, with lengths varying from 64 to 1024 frames.
• The UvA-*Nemo* dataset [9] is a facial analysis dataset that consists of 1240 videos. We apply the exact same pre-processing as for *VoxCeleb*. Each video starts with a neutral expression. Similar to Wang *et al.* [38], we use 1116 videos for training and 124 for evaluation.
• The *BAIR* robot pushing dataset [10] contains videos collected by a Sawyer robotic arm pushing diverse objects over a table. It consists of 42880 training and 128 test videos. Each video is 30 frame long and has a $256 \times 256$ resolution.
• Following Tulyakov *et al.* [34], we collected 280 tai-chi videos from YouTube. We use 252 videos for training and 28 for testing. Each video is split in short clips as described in pre-processing of *VoxCeleb* dataset. We retain only high quality videos and resized all the clips to $256 \times 256$ pixels (instead of $64 \times 64$ pixels in [34]). Finally, we obtain 3049 and 285 video chunks for training and testing respectively with video length varying from 128 to 1024 frames. This dataset is referred to as the *Tai-Chi-HD* dataset. The dataset will be made publicly available.

**Evaluation Protocol.** Evaluating the quality of image animation is not obvious, since ground truth animations are not available. We follow the evaluation protocol of Monkey-Net [28]. First, we

Table 1: Quantitative ablation study for video reconstruction on *Tai-Chi-HD*.

|  | $\mathcal{L}_1$ | Tai-Chi-HD (AKD, MKR) | AED |
|---|---|---|---|
| *Baseline* | 0.073 | (8.945, 0.099) | 0.235 |
| *Pyr.* | 0.069 | (9.407, 0.065) | 0.213 |
| *Pyr.*+$\mathcal{O}_{\mathbf{S}\leftarrow\mathbf{D}}$ | 0.069 | (8.773, 0.050) | 0.205 |
| *Jac. w/o Eq.* (12) | 0.073 | (9.887, 0.052) | 0.220 |
| *Full* | **0.063** | **(6.862, 0.036)** | **0.179** |

Table 2: Paired user study: user preferences in favour of our approach.

|  | X2Face [40] | Monkey-Net [28] |
|---|---|---|
| *Tai-Chi-HD* | 92.0% | 80.6% |
| *VoxCeleb* | 95.8% | 68.4% |
| *Nemo* | 79.8% | 60.6% |
| *Bair* | 95.0% | 67.0% |

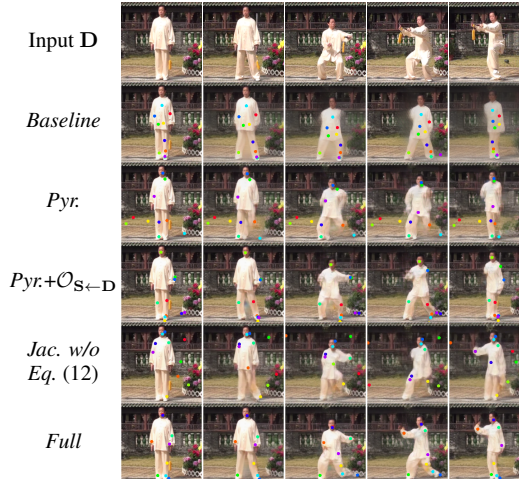

Input **D**

*Baseline*

*Pyr.*

*Pyr.*+$\mathcal{O}_{\mathbf{S}\leftarrow\mathbf{D}}$

*Jac. w/o Eq.* (12)

*Full*

Figure 3: Qualitative ablation on *Tai-Chi-HD*.

quantitatively evaluate each method on the "proxy" task of video reconstruction. This task consists of reconstructing the input video from a representation in which appearance and motion are decoupled. In our case, we reconstruct the input video by combining the sparse motion representation in (2) of each frame and the first video frame. Second, we evaluate our model on image animation according to a user-study. In all experiments we use $K$=10 as in [28]. Other implementation details are given in *Sup. Mat.*

**Metrics.** To evaluate video reconstruction, we adopt the metrics proposed in Monkey-Net [28]:

- $\mathcal{L}_1$. We report the average $\mathcal{L}_1$ distance between the generated and the ground-truth videos.
- *Average Keypoint Distance (AKD)*. For the *Tai-Chi-HD*, *VoxCeleb* and *Nemo* datasets, we use 3rd-party pre-trained keypoint detectors in order to evaluate whether the motion of the input video is preserved. For the *VoxCeleb* and *Nemo* datasets we use the facial landmark detector of Bulat *et al.* [5]. For the *Tai-Chi-HD* dataset, we employ the human-pose estimator of Cao *et al.* [7]. These keypoints are independently computed for each frame. AKD is obtained by computing the average distance between the detected keypoints of the ground truth and of the generated video.
- *Missing Keypoint Rate (MKR)*. In the case of *Tai-Chi-HD*, the human-pose estimator returns an additional binary label for each keypoint indicating whether or not the keypoints were successfully detected. Therefore, we also report the MKR defined as the percentage of keypoints that are detected in the ground truth frame but not in the generated one. This metric assesses the appearance quality of each generated frame.
- *Average Euclidean Distance (AED)*. Considering an externally trained image representation, we report the average euclidean distance between the ground truth and generated frame representation, similarly to Esser *et al.* [11]. We employ the feature embedding used in Monkey-Net [28].

**Ablation Study.** We compare the following variants of our model. *Baseline*: the simplest model trained without using the occlusion mask ($\mathcal{O}_{\mathbf{S}\leftarrow\mathbf{D}}$=1 in Eq. (8)), jacobians ($J_k = \mathbb{1}$ in Eq. (4)) and is supervised with $L_{rec}$ at the highest resolution only; *Pyr.*: the pyramid loss is added to *Baseline*; *Pyr.*+$\mathcal{O}_{\mathbf{S}\leftarrow\mathbf{D}}$: with respect to *Pyr.*, we replace the generator network with the occlusion-aware network; *Jac. w/o Eq.* (12) our model with local affine transformations but without equivariance constraints on jacobians Eq. (12); *Full*: the full model including local affine transformations described in Sec. 3.1.

In Fig. 3, we report the qualitative ablation. First, the pyramid loss leads to better results according to all the metrics except *AKD*. Second, adding $\mathcal{O}_{\mathbf{S}\leftarrow\mathbf{D}}$ to the model consistently improves all the metrics with respect to *Pyr.*. This illustrates the benefit of explicitly modeling occlusions. We found that without equivariance constraint over the jacobians, $J_k$ becomes unstable which leads to poor motion estimations. Finally, our *Full* model further improves all the metrics. In particular, we note that, with respect to the *Baseline* model, the MKR of the full model is smaller by the factor of 2.75. It shows that our rich motion representation helps generate more realistic images. These results are confirmed by our qualitative evaluation in Tab. 1 where we compare the *Baseline* and the *Full* models. In these experiments, each frame **D** of the input video is reconstructed from its first frame (first column) and the estimated keypoint trajectories. We note that the *Baseline* model does not locate any

Table 3: Video reconstruction: comparison with the state of the art on four different datasets.

| | Tai-Chi-HD | | | VoxCeleb | | | Nemo | | | Bair |
| | $\mathcal{L}_1$ | (AKD, MKR) | AED | $\mathcal{L}_1$ | AKD | AED | $\mathcal{L}_1$ | AKD | AED | $\mathcal{L}_1$ |
|---|---|---|---|---|---|---|---|---|---|---|
| X2Face [40] | 0.080 | (17.654, 0.109) | 0.272 | 0.078 | 7.687 | 0.405 | 0.031 | 3.539 | 0.221 | 0.065 |
| Monkey-Net [28] | 0.077 | (10.798, 0.059) | 0.228 | 0.049 | 1.878 | 0.199 | 0.018 | 1.285 | 0.077 | 0.034 |
| Ours | **0.063** | **(6.862, 0.036)** | **0.179** | **0.043** | **1.294** | **0.140** | **0.016** | **1.119** | **0.048** | **0.027** |

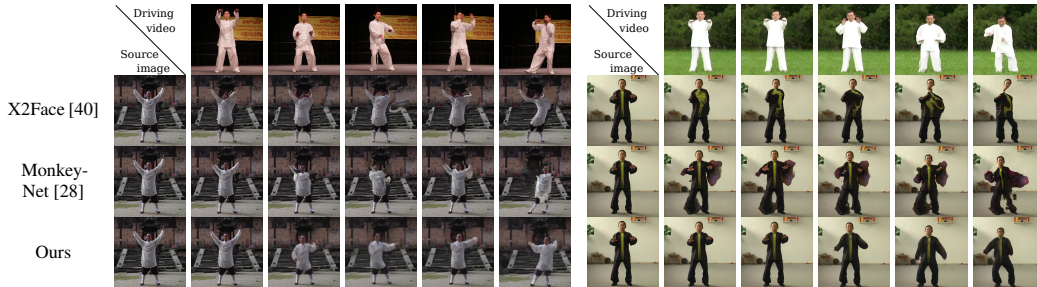

Figure 4: Qualitative comparison with state of the art for the task of image animation on two sequences and two source images from the *Tai-Chi-HD* dataset.

keypoints in the arms area. Consequently, when the pose difference with the initial pose increases, the model cannot reconstruct the video (columns 3,4 and 5). In contrast, the *Full* model learns to detect a keypoint on each arm, and therefore, to more accurately reconstruct the input video even in the case of complex motion.

**Comparison with State of the Art.** We now compare our method with state of the art for the video reconstruction task as in [28]. To the best of our knowledge, X2Face [40] and Monkey-Net [28] are the only previous approaches for model-free image animation. Quantitative results are reported in Tab. 3. We observe that our approach consistently improves every single metric for each of the four different datasets. Even on the two face datasets, *VoxCeleb* and *Nemo* datasets, our approach clearly outperforms X2Face that was originally proposed for face generation. The better performance of our approach compared to X2Face is especially impressive X2Face exploits a larger motion embedding (128 floats) than our approach (60=K*(2+4) floats). Compared to Monkey-Net that uses a motion representation with a similar dimension (50=K*(2+3)), the advantages of our approach are clearly visible on the *Tai-Chi-HD* dataset that contains highly non-rigid objects (*i.e.*human body).

We now report a qualitative comparison for image animation. Generated sequences are reported in Fig. 4. The results are well in line with the quantitative evaluation in Tab. 3. Indeed, in both examples, X2Face and Monkey-Net are not able to correctly transfer the body notion in the driving video, instead warping the human body in the source image as a blob. Conversely, our approach is able to generate significantly better looking videos in which each body part is independently animated. This qualitative evaluation illustrates the potential of our rich motion description. We complete our evaluation with a user study. We ask users to select the most realistic image animation. Each question consists of the source image, the driving video, and the corresponding results of our method and a competitive method. We require each question to be answered by 10 AMT worker. This evaluation is repeated on 50 different input pairs. Results are reported in Tab. 2. We observe that our method is clearly preferred over the competitor methods. Interestingly, the largest difference with the state of the art is obtained on *Tai-Chi-HD*: the most challenging dataset in our evaluation due to its rich motions.

## 5 Conclusions

We presented a novel approach for image animation based on keypoints and local affine transformations. Our novel mathematical formulation describes the motion field between two frames and is efficiently computed by deriving a first order Taylor expansion approximation. In this way, motion is described as a set of keypoints displacements and local affine transformations. A generator network combines the appearance of the source image and the motion representation of the driving video. In addition, we proposed to explicitly model occlusions in order to indicate to the generator network which image parts should be inpainted. We evaluated the proposed method both quantitatively and qualitatively and showed that our approach clearly outperforms state of the art on all the benchmarks.

## Footnotes

[1]https://github.com/AliaksandrSiarohin/first-order-model

[2]https://aliaksandrsiarohin.github.io/first-order-model-website/

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
