[Supplementary Material · 3854_sup.pdf]

# Supplementary material: First Order Motion Model for Image Animation

**Aliaksandr Siarohin**
DISI, University of Trento
aliaksandr.siarohin@unitn.it

**Stéphane Lathuilière**
DISI, University of Trento
LTCI, Télécom Paris, Institut polytechnique de Paris
stephane.lathuilire@telecom-paris.fr

**Sergey Tulyakov**
Snap Inc.
stulyakov@snap.com

**Elisa Ricci**
DISI, University of Trento
Fondazione Bruno Kessler
e.ricci@unitn.it

**Nicu Sebe**
DISI, University of Trento
Huawei Technologies Ireland
niculae.sebe@unitn.it

In this supplementary material, we provide derivations leading to the equations used in the main paper in Sec. A. Then, we provide implementation details in Sec. B. Finally, in Sec. C, we report further qualitative results of our method against the state of the art.

## A  Detailed Derivations

### A.1  Approximating Motion with Local Affine Transformations

Here, we detail the derivation leading to the approximation of $\mathcal{T}_{\mathbf{S}\leftarrow\mathbf{D}}$ near the keypoint $z_k$ in Eq. (4) of the main paper. Using first order Taylor expansion we can obtain:

$$\mathcal{T}_{\mathbf{S}\leftarrow\mathbf{D}}(z) = \mathcal{T}_{\mathbf{S}\leftarrow\mathbf{D}}(z_k) + \left(\frac{d}{dz}\mathcal{T}_{\mathbf{S}\leftarrow\mathbf{D}}(z)\Big|_{z=z_k}\right)(z-z_k) + o(\|z-z_k\|) \tag{1}$$

$\mathcal{T}_{\mathbf{S}\leftarrow\mathbf{D}}$ can be written as the composition of two transformations:

$$\mathcal{T}_{\mathbf{S}\leftarrow\mathbf{D}} = \mathcal{T}_{\mathbf{S}\leftarrow\mathbf{R}} \circ \mathcal{T}_{\mathbf{R}\leftarrow\mathbf{D}} \tag{2}$$

In order to compute the zeroth order term, we estimate the transformation $\mathcal{T}_{\mathbf{R}\leftarrow\mathbf{D}}$ near the point $z_k$ in the driving frame $\mathbf{D}$, e.g $p_k = \mathcal{T}_{\mathbf{R}\leftarrow\mathbf{D}}(z_k)$. Then we can estimate the transformation $\mathcal{T}_{\mathbf{S}\leftarrow\mathbf{R}}$ near $p_k$ in the reference $\mathbf{R}$. Since $p_k = \mathcal{T}_{\mathbf{R}\leftarrow\mathbf{D}}(z_k)$ and $\mathcal{T}_{\mathbf{R}\leftarrow\mathbf{D}}^{-1} = \mathcal{T}_{\mathbf{D}\leftarrow\mathbf{R}}$, we can write $z_k = \mathcal{T}_{\mathbf{D}\leftarrow\mathbf{R}}(p_k)$. Consequently, we obtain:

$$\begin{aligned}
\mathcal{T}_{\mathbf{S}\leftarrow\mathbf{D}}(z_k) &= \mathcal{T}_{\mathbf{S}\leftarrow\mathbf{R}} \circ \mathcal{T}_{\mathbf{R}\leftarrow\mathbf{D}}(z_k) \\
&= \mathcal{T}_{\mathbf{S}\leftarrow\mathbf{R}} \circ \mathcal{T}_{\mathbf{D}\leftarrow\mathbf{R}}^{-1}(z_k) \\
&= \mathcal{T}_{\mathbf{S}\leftarrow\mathbf{R}} \circ \mathcal{T}_{\mathbf{D}\leftarrow\mathbf{R}}^{-1} \circ \mathcal{T}_{\mathbf{D}\leftarrow\mathbf{R}}(p_k) \\
&= \mathcal{T}_{\mathbf{S}\leftarrow\mathbf{R}}(p_k).
\end{aligned} \tag{3}$$

Concerning the first order term, we apply the function composition rule in Eq. (2) and obtain:

$$\left(\frac{d}{dz}\mathcal{T}_{\mathbf{S}\leftarrow\mathbf{D}}(z)\Big|_{z=z_k}\right) = \left(\frac{d}{dp}\mathcal{T}_{\mathbf{S}\leftarrow\mathbf{R}}(p)\Big|_{p=\mathcal{T}_{\mathbf{R}\leftarrow\mathbf{D}}(z_k)}\right)\left(\frac{d}{dz}\mathcal{T}_{\mathbf{D}\leftarrow\mathbf{R}}^{-1}(z)\Big|_{z=z_k}\right) \tag{4}$$

Since the matrix inverse of the Jacobian is equal to the Jacobian of the inverse function, and since $p_k = \mathcal{T}_{\mathbf{R}\leftarrow\mathbf{D}}(z_k)$, Eq. (4) can be rewritten:

$$\left(\frac{d}{dz}\mathcal{T}_{\mathbf{S}\leftarrow\mathbf{D}}(z)\Big|_{z=z_k}\right) = \left(\frac{d}{dp}\mathcal{T}_{\mathbf{S}\leftarrow\mathbf{R}}(p)\Big|_{p=p_k}\right)\left(\frac{d}{dp}\mathcal{T}_{\mathbf{D}\leftarrow\mathbf{R}}(p)\Big|_{p=p_k}\right)^{-1} \tag{5}$$

After injecting Eqs. (3) and (5) into (1), we finally obtain:

$$\mathcal{T}_{\mathbf{S}\leftarrow\mathbf{D}}(z) \approx \mathcal{T}_{\mathbf{S}\leftarrow\mathbf{R}}(p_k) + \left(\frac{d}{dp}\mathcal{T}_{\mathbf{S}\leftarrow\mathbf{R}}(p)\Big|_{p=p_k}\right)\left(\frac{d}{dp}\mathcal{T}_{\mathbf{D}\leftarrow\mathbf{R}}(p)\Big|_{p=p_k}\right)^{-1}(z - \mathcal{T}_{\mathbf{D}\leftarrow\mathbf{R}}(p_k)) \quad (6)$$

## A.2 Equivariance Loss

At training time, we use equivariance constraints that enforces:

$$\mathcal{T}_{\mathbf{X}\leftarrow\mathbf{R}} \equiv \mathcal{T}_{\mathbf{X}\leftarrow\mathbf{Y}} \circ \mathcal{T}_{\mathbf{Y}\leftarrow\mathbf{R}} \quad (7)$$

After applying first order Taylor expansion on the left-hand side, we obtain:

$$\mathcal{T}_{\mathbf{X}\leftarrow\mathbf{R}}(p) = \mathcal{T}_{\mathbf{X}\leftarrow\mathbf{R}}(p_k) + \left(\frac{d}{dp}\mathcal{T}_{\mathbf{X}\leftarrow\mathbf{R}}(p)\Big|_{p=p_k}\right)(p - p_k) + o(\|p - p_k\|). \quad (8)$$

After applying first order Taylor expansion on the right-hand side in Eq. (7), we obtain:

$$\mathcal{T}_{\mathbf{X}\leftarrow\mathbf{Y}}\circ\mathcal{T}_{\mathbf{Y}\leftarrow\mathbf{R}}(p) = \mathcal{T}_{\mathbf{X}\leftarrow\mathbf{Y}}\circ\mathcal{T}_{\mathbf{Y}\leftarrow\mathbf{R}}(p_k) + \left(\frac{d}{dp}\mathcal{T}_{\mathbf{X}\leftarrow\mathbf{Y}}\circ\mathcal{T}_{\mathbf{Y}\leftarrow\mathbf{R}}\Big|_{p=p_k}\right)(p - p_k) + o(\|p - p_k\|), \quad (9)$$

We can further simplify this expression using derivative of function composition:

$$\left(\frac{d}{dp}\mathcal{T}_{\mathbf{X}\leftarrow\mathbf{Y}}\circ\mathcal{T}_{\mathbf{Y}\leftarrow\mathbf{R}}\Big|_{p=p_k}\right) = \left(\frac{d}{dp}\mathcal{T}_{\mathbf{X}\leftarrow\mathbf{Y}}(p)\Big|_{p=\mathcal{T}_{\mathbf{Y}\leftarrow\mathbf{R}}(p_k)}\right)\left(\frac{d}{dp}\mathcal{T}_{\mathbf{Y}\leftarrow\mathbf{R}}(p)\Big|_{p=p_k}\right). \quad (10)$$

Eq. (7) holds only when every coefficient in Taylor expansion of the right and left sides are equal. Thus, it leads us to the following constraints:

$$\mathcal{T}_{\mathbf{X}\leftarrow\mathbf{R}}(p_k) \equiv \mathcal{T}_{\mathbf{X}\leftarrow\mathbf{Y}}\circ\mathcal{T}_{\mathbf{Y}\leftarrow\mathbf{R}}(p_k), \quad (11)$$

and

$$\left(\frac{d}{dp}\mathcal{T}_{\mathbf{X}\leftarrow\mathbf{R}}(p)\Big|_{p=p_k}\right) \equiv \left(\frac{d}{dp}\mathcal{T}_{\mathbf{X}\leftarrow\mathbf{Y}}(p)\Big|_{p=\mathcal{T}_{\mathbf{Y}\leftarrow\mathbf{R}}(p_k)}\right)\left(\frac{d}{dp}\mathcal{T}_{\mathbf{Y}\leftarrow\mathbf{R}}(p)\Big|_{p=p_k}\right). \quad (12)$$

## A.3 Transferring Relative Motion

In order to transfer only relative motion patterns, we propose to estimate $\mathcal{T}_{\mathbf{S}_t\leftarrow\mathbf{R}}(p)$ near the keypoint $p_k$ by shifting the motion in the driving video to the location of keypoint $p_k$ in the source. To this aim, we introduce $\mathcal{V}_{\mathbf{S}_1\leftarrow\mathbf{D}_1}(p_k) = \mathcal{T}_{\mathbf{S}_1\leftarrow\mathbf{R}}(p_k) - \mathcal{T}_{\mathbf{D}_1\leftarrow\mathbf{R}}(p_k) \in \mathbb{R}^2$ that is the 2D vector from the landmark position $p_k$ in $\mathbf{D}_1$ to its position in $\mathbf{S}_1$. We proceed as follows. First, we shift point coordinates according to $-\mathcal{V}_{\mathbf{S}_1\leftarrow\mathbf{D}_1}(p_k)$ in order to obtain coordinates in $\mathbf{D}_1$. Second, we apply the transformation $\mathcal{T}_{\mathbf{D}_t\leftarrow\mathbf{D}_1}$. Finally, we translate the points back in the original coordinate space using $\mathcal{V}_{\mathbf{S}_1\leftarrow\mathbf{D}_1}(p_k)$. Formally, it can be written:

$$\mathcal{T}_{\mathbf{S}_t\leftarrow\mathbf{R}}(p) = \mathcal{T}_{\mathbf{D}_t\leftarrow\mathbf{D}_1}\big(\mathcal{T}_{\mathbf{S}_1\leftarrow\mathbf{R}}(p) - \mathcal{V}_{\mathbf{S}_1\leftarrow\mathbf{D}_1}(p_k)\big) + \mathcal{V}_{\mathbf{S}_1\leftarrow\mathbf{D}_1}(p_k)$$

Now, we can compute the value and Jacobian in the $p_k$:

$$\mathcal{T}_{\mathbf{S}_t\leftarrow\mathbf{R}}(p_k) = \mathcal{T}_{\mathbf{D}_t\leftarrow\mathbf{D}_1}\circ\mathcal{T}_{\mathbf{D}_1\leftarrow\mathbf{R}}(p_k) - \mathcal{T}_{\mathbf{D}_1\leftarrow\mathbf{R}}(p_k) + \mathcal{T}_{\mathbf{S}_1\leftarrow\mathbf{R}}(p_k)$$

and:

$$\left(\frac{d}{dp}\mathcal{T}_{\mathbf{S}_t\leftarrow\mathbf{R}}(p)\Big|_{p=p_k}\right) = \left(\frac{d}{dp}\mathcal{T}_{\mathbf{D}_t\leftarrow\mathbf{R}}(p)\Big|_{p=p_k}\right)\left(\frac{d}{dp}\mathcal{T}_{\mathbf{D}_1\leftarrow\mathbf{R}}(p)\Big|_{p=p_k}\right)^{-1}\left(\frac{d}{dp}\mathcal{T}_{\mathbf{S}_1\leftarrow\mathbf{R}}(p)\Big|_{p=p_k}\right).$$

Now using Eq. (6) and treating $\mathbf{S}_1$ as source and $\mathbf{S}_t$ as driving frame, we obtain:

$$\mathcal{T}_{\mathbf{S}_1\leftarrow\mathbf{S}_t}(z) \approx \mathcal{T}_{\mathbf{S}_1\leftarrow\mathbf{R}}(p_k) + J_k(z - \mathcal{T}_{\mathbf{S}\leftarrow\mathbf{R}}(p_k) + \mathcal{T}_{\mathbf{D}_1\leftarrow\mathbf{R}}(p_k) - \mathcal{T}_{\mathbf{D}_t\leftarrow\mathbf{R}}(p_k)) \quad (13)$$

with

$$J_k = \left(\frac{d}{dp}\mathcal{T}_{\mathbf{D}_1\leftarrow\mathbf{R}}(p)\Big|_{p=p_k}\right)\left(\frac{d}{dp}\mathcal{T}_{\mathbf{D}_t\leftarrow\mathbf{R}}(p)\Big|_{p=p_k}\right)^{-1}. \quad (14)$$

Note that, here, $\left(\frac{d}{dp}\mathcal{T}_{\mathbf{S}_1\leftarrow\mathbf{R}}(p)\Big|_{p=p_k}\right)$ canceled out.

# B    Implementation details

## B.1    Architecture details

In order to reduce memory and computational requirements of our model, the keypoint detector and dense motion predictor both work on resolution of $64 \times 64$ (instead of $256 \times 256$). For the two networks of the motion module, we employ an architecture based on U-Net [3] with five $conv_{3\times3}$ - $bn$ - $relu$ - $avg - pool_{2\times2}$ blocks in the encoders and five $upsample_{2\times2}$ - $conv_{3\times3}$ - $bn$ - $relu$ blocks in the decoders. In the generator network, we use the Johnson architecture [1] with two down-sampling blocks, six residual-blocks and two up-sampling blocks. We train our network using Adam [2] optimizer with learning rate $2e - 4$ and batch size 20. We employ learning decay by dropping the learning rate at $\frac{T}{2}$ and $\frac{3T}{4}$ iterations, where T is total number of iteration. We chose $T \approx 100k$ for *Tai-Chi-HD* and *VoxCeleb*, and $T \approx 40k$ for *Nemo* and *Bair*. The model converges in approximately 2 days using 2 TitanX gpus for *Tai-Chi-HD* and *VoxCeleb*.

## B.2    Equivariance loss implementation

As explained above our equivariance losses force the keypoint detector to be equivariant to some transformations $\mathcal{T}_{\mathbf{X}\leftarrow\mathbf{Y}}$. In our experiments $\mathcal{T}_{\mathbf{X}\leftarrow\mathbf{Y}}$ is implemented using randomly sampled thin plate splines. We sample spline parameters from normal distributions with zero mean and variance equal to 0.005 for deformation component and 0.05 for the affine component. For deformation component we use uniform $5 \times 5$ grid.

# C    Additional experiments

## C.1    Image Animation

In this section, we report additional qualitative results.

We compare our approach with X2face [5] and Monkey-Net [4]. In Fig. 1, we show three animation examples from the *VoxCeleb* dataset. First, X2face is not capable of generating realistic video sequences as we can see, for instance in the last frame of the last sequence. Then, Monkey-Net generates realistic frames but fails to generate specific facial expressions as in the third frame of the first sequence or in transferring the eye movements as in the last two frames of the second sequence.

In Fig. 2, we show three animation examples from the *Nemo* dataset. First, we observe that this dataset is simpler than *VoxCeleb* since the persons are facing a uniformly black background. With this simpler dataset, X2Face generates realistic videos. However, it is not capable of inpainting image parts that are not visible in the source image. For instance, X2Face does not generate the teeth. Our approach also perform better than Monkey-Net as we can see by comparing the generate teeth in the first sequence or the closed eyes in the fourth frames of the second and third sequences.

In Fig. 2, we report additional examples for the *Tai-Chi-HD* dataset. These examples are well in line with what is reported in the main paper. Both X2Face and Monkey-Net completely fail to generate realistic videos. The source images are warped without respecting human body structure. Conversely, our approach is able to deform the person in foreground without affecting the background. Even though we can see few minor artifacts, our model is able to move each body part independently following the body motion in the driving video.

Finally, in Fig. 4 we show three image animation examples on the *Bair* dataset. Again, we see that X2Face is not able to transfer motion since it constantly returns frames almost identical with the source images. Compared to Monkey-Net, our approach performs slightly better since it preserves better the robot arm as we can see in the second frame of the first sequence or in the fourth frame of the last sequence.

## C.2    Keypoint detection

We now illustrate the keypoints that are learned by our self-supervised approach in Fig. 5. On the *Tai-Chi-HD* dataset, the keypoints are semantically consistent since each of them corresponds to a body part: light green for the right foot, and blue and red for the face for instance. Note that, a light

green keypoint is constantly located in the bottom left corner in order to model background or camera motion. On *VoxCeleb*, we observe that, overall, the obtained keypoints are semantically consistent except for the yellow and green keypoints. For instance, the red and purple keypoints constantly correspond to the nose and the chin respectively. We observe a similar consistency for the *Nemo* dataset. For the Bair dataset, we note that two keypoints (dark blue and light green) correspond to the robotic arm.

### C.3  Visualizing occlusion masks

In Fig. 6, we visualize the predicted occlusion masks $\hat{\mathcal{O}}_{\mathbf{S}\leftarrow\mathbf{D}}$ on the *Tai-Chi-HD*, *VoxCeleb* and *Nemo* datasets. In the first sequence, when the person in the driving video is moving backward (second to fourth frames), the occlusion mask becomes black (corresponding to 0) in the background regions that are occluded in the source frame. It indicates that these parts cannot be generated by warping the source image features and must be inpainted. A similar observation can be made on the example sequence of *VoxCeleb*. Indeed, we see that when the face is rotating, the mask has low values (dark grey) in the neck region and in the right face side (in the left-hand side of the image) that are not visible in the source Frame. Then, since the driving video example from *Nemo* contains only little motion, the predicted mask is almost completely white. Overall, these three examples show that the occlusion masks truly indicate occluded regions even if no specific training loss is employed in order to lead to this behaviour. Finally, the predicted occlusion masks are more difficult to interpret in the case of the *Bair* dataset. Indeed, the robotic arm is masked out in every frame whereas we could expect that the model generates it by warping. A possible explanation is that, since in this particular dataset, the moving object is always the same, the network can generate without warping the source image. We observe also that masks have low values for the regions corresponding to the arm shadow. It is explained by the fact that shadows cannot be obtained by image warping and that they need to be added by the generator.

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

Figure 1: Qualitative comparison with state of the art for the task of image animation on different sequences from the *VoxCeleb* dataset.

Figure 2: Qualitative comparison with state of the art for the task of image animation on different sequences from the *Nemo* dataset.

Figure 3: Qualitative comparison with state of the art for the task of image animation on different sequences from the *Tai-Chi-HD* dataset.

Figure 4: Qualitative comparison with state of the art for the task of image animation on different sequences from the *Bair* dataset.

Figure 5: Keypoint visualization for the four datasets.

Figure 6: Visualization of occlusion masks and images obtained after deformation on *Tai-Chi-HD*, *VoxCeleb*, *Nemo* and *Bair* datasets.