[Reviews · NeurIPS 2019]

Reviewer 1



Summary: The system attacks the problem of generating images that conform to a given source image driven by motion estimated from a given video. To that end, the system estimates sparse transformations between in terms of corresponding key points and, different from [21], the local linear deformations around those key points. These transformations are composed from transformations with respect to a common reference configuration. The sparse transformations are converted with a CNN into dense motion and occlusion masks. Finally, the motion and occlusion are combined by another neural network with the input image to create the final output. Positive: The paper introduces the novel idea of first order motion and occlusion modeling to unsupervised image animation. Apart from the basic training idea, which is an extension of the equivariance constraints, the authors need to simplify the transformations learned by the networks by performing a number of well chosen geometric tricks (e.g. line 164, 184) or mapping variations (e.g. using transformations D_t<-D_1 instead of D_t<-S). The mathematics of the first order motion are well described, and the supplementary material covers well the missing details in the manuscript. The authors show good performance in three very different applications (faces, full bodies, robots), where they clearly outperform state-of-the-art Negative: My main complain about the paper is that understanding some aspects from it is hard, and could use a description of the intuitions behind them. For example: - what is the intuition between the difference of gaussians used in the heatmap H_k (section 2.3 supp mat)? - How do the reference configurations R look like? Is there any intuition behind those configurations? Could they be visualized (in supp mat obviously)? - Related to the equivariance constraint, could it be said that it encourages the key points to segment the object so that it deforms as close as possible to the thin plate splines model? This holds also to the affine model around the joints, which should conform to the thin plate splines model, right? A secondary concern is the reproducibility of the system. Its large number of components would make implementing it from scratch very hard. I think the value of the paper would increase if the authors release the source code. I found a couple of typos: - Forth->Fourth, line 42 - Latter -> later, line 95

Reviewer 2



This paper proposed a first order motion model for image animation. Firstly, motion is modeled as a set of keypoints and local affine transformations, which has the capability to model large object pose changes compared to the previously proposed zeroth order model. This paper approximated the motion between two frames based on first order Taylor expansions. Secondly, this paper proposed to model occlusions to indicate the generator network about which image parts should be inpainted. This is needed to handle the large motion patterns in the driving video. Pros: - Extensive experiments are performed on high resolution datasets. The proposed method shows clear improvement over the previous methods quantitatively and qualitatively. - This paper also released a new high resolution dataset Thai-Chi-HD for the community. Cons: - Lack of insight into why the first order motion model is necessary. For example, in the ablation test section, I see a gap between "Full" model and "Pyr.+O" model. It would be nice to include a test showing all the factors in between, e.g. first order model vs zeroth order model. - Missing citations for GANs and VAEs in line 24-25. I. Goodfellow, J. Pouget-Abadie, M. Mirza, B. Xu, D. Warde-Farley, S. Ozair, A. Courville, and Y. Bengio. Generative adversarial nets. In NIPS, 2014. D. P. Kingma and M. Welling. Auto-encoding variational bayes. In ICLR, 2014. - Clarity improvements: Reconstruction loss Line 194: which VGG-19 layer did you use exactly? What are the resolutions used for reconstruction loss? Instead of a pyramid of resolutions, have you tried to compute the same loss on multiple layers of the same input resolution? In such a way, it's more efficient as only one feedforward path is needed. A few minor issues: Line 243: How do you preserve the aspect ratio if you resize a video to the fixed target 256x256 resolution? Line 172: is moves -> is moved Line 196: similarly to -> similar to Line 203: Lets -> Let's Line 240: until its it -> until it is Line 302: every single metrics -> every single metric

Reviewer 3



Originality: 1) The task of image animation is not new, but the proposed motion model is novel. 2) The clear difference between this paper and previous papers lies in the first order motion model. 3) Related work is adequately cited and compared. Quality: 1) The submission is technically sound. The mathematical proof seems correct to me. 2) This is a complete piece of work. The authors conduct extensive experiments on several benchmark datasets to validate the effectiveness of the proposed first order motion model. 3) Failure case analysis is missing. 4) What is the intuition of using first order motion model? Can the authors explain how this connects to affine transformation? Can we use a second order motion model? 5) What is the virtual reference frame? Does it represent a canonical pose for a specific object category? Clarity: The method part is a bit unclear. 1) It seems to me that the zeroth order motion is the keypoint location. However, in Figure 1, T_{S<-D}(p_k) seems to be pixel displacement, which is different from the keypoint location. 2) Can the authors explain L161-L163 in more details? Significance: This paper address a difficult problem in a better way than previous papers. It demonstrably advances the state-of-the-art. I believe it provides a good baseline for the following researchers to build upon and make significant progress.

[Author Response · NeurIPS 2019]

We thank the reviewers for their valuable comments. We will add the suggested citations and fix the typos and update
the manuscript and the supplement with the discussions below. To ensure reproducibility, we will release the code and
trained models. We now address all the comments individually.

**R1-3. First-order model.** The equation $\mathcal{T}_{\mathbf{S}\leftarrow\mathbf{D}}(z) \approx \mathcal{T}_{\mathbf{S}\leftarrow\mathbf{R}}(p_k) + J_k(z - \mathcal{T}_{\mathbf{D}\leftarrow\mathbf{R}}(p_k))$ (4) defines how the coordinates
in the neighbourhood of the keypoint should be transformed. Zeroth-order approximation for $\mathcal{T}_{\mathbf{S}\leftarrow\mathbf{D}}(z)$ is $\mathcal{T}_{\mathbf{S}\leftarrow\mathbf{R}}(p_k)$
which does not produce meaningful motion since all the points in a neighborhood of a keypoint are mapped to the
same location. Instead, Monkey-net [21] considers zeroth-order approximation for $\mathcal{T}_{\mathbf{S}\leftarrow\mathbf{D}}(z) - z$, which corresponds
to $J_k = \mathbb{1}$ in Eq.(4). It assumes that motion is constant in the neighbourhood of keypoints (pixels are simply shifted).
We call this a zeroth-order model. Such model, for example, cannot handle objects moving towards the camera. In
contrast, the proposed first-order model can handle such motion by introducing $J_k$, which models the corresponding
affine transformation.

**R1-3. Reference frame.** To define $J_k$ we assume there exists an abstract reference frame which is an arbitrary
representation of the object. We note that the reference frame is an abstract concept that cancels out in our derivations,
therefore not allowing us to visualize it. Then, Eq.(4) can be viewed as mapping the object from the driving frame to
this arbitrary representation, followed by mapping it to the source frame. Note that unlike [31] there is no dependence
on the reference frame in the final model, the only constraint we have on the reference is that it should be similar for
both source and driving frames near keypoints locations. In fact as long as $p_k$ (keypoint coordinates in the reference
frame) is the same for source and driving keypoints, we can move the object to an arbitrary location in the reference
frame without changing the target result. Therefore, the coordinate of $p_k$ cannot be estimated and visualized.

**R1. Difference of the gaussian heatmaps.** Gaussian heatmaps are usually employed for pose guided generation [2].
Here, we use the difference of gaussians to indicate to the dense motion predictor where the source keypoints are.
This information helps predict occlusion maps. Concatenation of source and driving heatmaps requires twice as much
channels and hence is computationally less efficient. A similar representation was used in [21].

**R1. Equivariance constraints.** Since there is no direct supervision on the keypoints, their predictions can be
unstable. Equivariance constraint forces the model to predict consistent keypoints with respect to a known geometric
transformation $\mathcal{T}_{\mathbf{X}\leftarrow\mathbf{Y}}$. We used thin plate splines as they have been previously employed in unsupervised keypoint
detection [15, 32] and are similar to natural image deformations. Similarly to keypoints locations we introduce
equivariance constraints for jacobians in the keypoint neighbourhood.

**R2. Gap between "Full" and "Pyr.+O" model.** The difference between Full and Pyr.+O corresponds exactly to the
addition of Jacobians and of the equivariance loss over Jacobians, e.g Eq.(11). Note that Pyr.+O and other baselines use
equivariance loss over keypoint locations, e.g Eq.(10). If we disable equivariance loss over Jacobians they will become
unstable, and we get the following results $\mathcal{L}_1$ - 0.073, (AKD, MKR) - (9.89, 0.052), AED - 0.22. Note this performance
is significantly worse than Pyr.+O, and similar to the Baseline (see Tab. 1).

**R2. Reconstruction loss.** We use the following layers for each resolution conv1_2, conv2_2, conv3_2, conv4_2,
conv5_2, similarly to [28]. The resolutions are $256 \times 256$, $128 \times 128$, $64 \times 64$ and $32 \times 32$. There are $5 \times 4$ terms in
total. The pyramid loss implementation suggested by R2 corresponds to the loss used by [28], which in our experiments
led to very blurry results (see Fig. 2). Our loss adds minimal computational overhead compared to the loss of [28].

**R2. Aspect ratio.** We cropped faces using square bounding boxes. If the detector outputs a rectangular bounding box
of size $300 \times 200$, we first enlarge this bounding box to $300 \times 300$ and then crop. Thus, the aspect ratio does not change.

**R3. Failure cases.** Failure cases will be added to *Supp.Mat.* and to the associated video. Overall, failures cases are
often due to not realistic inpainting (Example 1,3 and 5 in the figure below) or rare motions, underrepresented in training
data (Example 2 and 4 in the figure below). Zoom-in for greater detail.

**R3. Second order motion model.** Second order extension of our model is possible but is a subject of future work.It
will require computation of the second derivative of the inverse of $\mathcal{T}$ which is a three dimensional tensor.

**R3. Fig. 1.** The intuition behind Fig. 1 is that we combine keypoint displacement from motion model of [21] (see
above), with $J_k$. We will clarify this in the camera ready.

**R3. Clarifications about L161-L163.** ConvNets are known to have difficulties to use information from input regions
to generate spatially distant output regions. For example, it is hard for a CNN to utilize information from a region on
the left to generate a region on the right. To simplify this task, similarly to [21] we warp the images such that the input
regions in the source image have approximately the same coordinates as their output counterparts. For more information
see [21] Sections 3.3-3.4.

[Meta-Review · NeurIPS 2019]

As all reviewers agreed (especially after taking into account the authors' responses) this is a solid paper presented an interesting method for image animation. For the final version please address all reviewers' comments.